# Morphological and Optical Characterization of Colored Nanotubular Anodic Titanium Oxide Made in an Ethanol-Based Electrolyte

**DOI:** 10.3390/ma14226992

**Published:** 2021-11-18

**Authors:** Marta Michalska-Domańska, Mateusz Czerwiński, Magdalena Łazińska, Vikas Dubey, Marcin Jakubaszek, Zbigniew Zawadzki, Jerzy Kostecki

**Affiliations:** 1Institute of Optoelectronics, Military University of Technology, 2 Kaliskiego Str., 00-908 Warsaw, Poland; mateusz.czerwinski@wat.edu.pl (M.C.); marcin.jakubaszek@wat.edu.pl (M.J.); zbigniew.zawadzki@wat.edu.pl (Z.Z.); jerzy.kostecki@wat.edu.pl (J.K.); 2Institute of Materials Science, Military University of Technology, 2 Kaliskiego Str., 00-908 Warsaw, Poland; magdalena.lazinska@wat.edu.pl; 3Department of Physics, Bhilai Institute of Technology, Raipur 493661, India; jsvikasdubey@gmail.com

**Keywords:** titanium anodization, anodic titanium oxide, ethanol-based electrolyte, reflectance, Commission internationale de l’éclairage (CIE) color space, colored oxide, structural coloring

## Abstract

In this paper, the possibility of color controlling anodic titanium oxide by changing anodizing conditions of titanium in an ethanol-based electrolyte is demonstrated. Colored anodic titanium oxide was fabricated in an ethanol-based electrolyte containing 0.3 M ammonium fluoride and various amounts of deionized water (2, 3.5, 5, or 10 vol%), at voltages that varied from 30 to 60 V and at a constant anodization temperature of 20 °C. Morphological characterization of oxide layers was established with the use of a scanning electron microscope. Optical characterization was determined by measuring diffusion reflectance and calculating theoretical colors. The resulting anodic oxides in all tested conditions had nanotubular morphology and a thickness of up to hundreds of nanometers. For electrolytes with 3.5, 5, and 10 vol% water content, the anodic oxide layer thickness increased with the applied potential increase. The anodic titanium oxide nanotube diameters and the oxide thickness of samples produced in an electrolyte with 2 vol% water content were independent of applied voltage and remained constant within the error range of all tested potentials. Moreover, the color of anodic titanium oxide produced in an electrolyte with 2 vol% of water was blue and was independent from applied voltage, while the color of samples from other electrolyte compositions changed with applied voltage. For samples produced in selected conditions, iridescence was observed. It was proposed that the observed structural color of anodic titanium oxide results from the synergy effect of nanotube diameter and oxide thickness.

## 1. Introduction

Structural color is widespread in the biological world and can be observed, for example, in the feathers of birds [1] or the wings of insects [2]. This phenomenon is due to interactions of natural light with periodic microstructures or nanostructures present on surfaces that have geometrical features and sizes comparable to visible wavelengths.

Anodization is a fast and relatively inexpensive method of producing nanostructured oxide on the tops of valve metals (such as Al, Ti, Zn, Sn, W, etc. [3,4,5,6,7]) and their alloys (such as Ti6Al7Nb, FeAl, etc. [8,9,10,11,12]). The most popular metals used in anodization are aluminum and titanium. Anodic aluminum oxide (AAO) is well arranged and perpendicular to the Al substrate nanocapillars [13]. Generally, anodic aluminum oxide is used in two main applications: as a protective, anticorrosion layer on the tops of aluminum alloys [14,15], and as a template for nanofabrication when the nanocapillars are hexagonally arranged [16,17]. 

Another important nanostructured anodic material is anodic titanium oxide (ATO) [18], which can be obtained in two basic morphologies: nanoporous ATO (which is similar to AAO, but with a significantly poorer nanopore arrangement) or nanotubular ATO [19,20]. Generally, ATO is used in photocatalytic applications to achieve energy conservation and environment protection [21,22]. In addition, AAO and ATO can be used in many other applications, such as providing a substrate for surface-enhanced Raman spectroscopy [23,24], implants [25,26], or coloring metal surfaces [27,28]. 

Generally, the growing of nanometric oxide films at the metal surface exhibits entrancing colors due to their interference with light. This effect can be used, for example, in architecture [28] or optoelectronics [29]. When no pigments are associated with generation of color surface, the observed colors are known as interference colors. The observed color comes from periodic nanostructures covering the surface and can be caused by the nature of nanostructures (the size of nano objects; nano objects’ arrangements or lack thereof; or stoichiometric defects in oxide layer composition), or by the thickness of layers with nanostructures [27,29]. 

Colored oxide can be fabricated on an aluminum or titanium surface by anodization, wherein applied anodizing conditions strongly affect the resulting colors and their intensities [30]. Commercial pure titanium alloy, anodized in sulfuric acid electrolyte at the potential range of 5–80 V for 30 s, leads to the formation of anodic oxide with color, depending on applied potential [27]. Diamanti et al. analyzed in detail the influence of sulfuric acid concentration in an electrolyte used in one-step anodizing of titanium [31]. The interference color of anodic oxide on commercially pure titanium and Ti-6Al-4V was fabricated with success by one-step anodizing in a sulfuric acid electrolyte or by two-step anodizing, with hydrochloric acid in the first anodizing step and phosphoric acid in the second [32]. Titanium anodic oxide made in an electrolyte containing 3.5 M HCl solve in 2-propanol, at the potential of 15 V for different times (5–60 min), was characterized by colors ranging from violet to blue, depending on the anodizing time [33]. As discussed, the color of ATO is affected by anodic oxide thickness, which is controlled by anodizing applied voltage [28,34,35]. Moreover, since Pedeferri patented an electrochemical method for the coloration of titanium surfaces in 2002 [36], this method has become very commonly used in art [28].

In addition to the focus on colored anodic titanium oxide, colored anodic aluminum oxide was investigated. For example, colored AAO was successfully fabricated by anodization of Al in oxalic acid [37], phosphoric acid [38,39,40], sulfuric acid [41], mixed organic acidic electrolyte (citric mixed with malic acid, with the addition of propylene glycol) [42], etidronic acid [43], and/or alkaline electrolyte (KOH, Na_4_P_2_O_7_·10H_2_O, and Na_2_SiO_3_) [44]. It was shown that the structural color of oxide depends on anodizing conditions, especially the electrolyte type [38,42], the applied voltage or current density [43,44], and the anodizing time [38,39], all of which are strongly related to oxide thickness and nanopore diameter. Importantly, it is possible to increase the saturation of the demonstrated color of AAO by adding an outer metallic coating with defined thickness or roughness [44]. 

Zhang et al. produced colored AAO in 0.6 M phosphoric acid modified by propylene glycol at 100 V and at −5 °C, demonstrating that a red-shift appears in the reflection spectrum of produced AAO samples when anodization time increases [40]. In addition, they demonstrated that the Au-coated AAO films displayed highly saturated colors when compared with uncoated samples [40]. Liu et al. fabricated colored AAO by Al anodization conducted in 0.3 M sulfuric acid at 3 °C in the voltage range of 10–25 V [41]. Moreover, they used, with success, prepared AAO oxide films covered by a layer of Ag as a colorimetric sensor for nitroaromatic detection [41]. The changes in layered AAO color, based on oxide thickness and/or nanopore diameter and arrangements made on each oxide layer, were investigated [45,46,47,48]. In this way, the photonic crystals characterized by very bright structural colors were fabricated [46,48].

As discussed, colored anodic oxides are still a fresh research topic, especially in relation to anodic titanium oxide. The demand for colored titanium surfaces has increased because of their many applications, including decoration and architecture [28]. As pigments used to produce colored surfaces often contain heavy and/or harmful elements, new approaches to coloring metal surfaces without pigments are desirable. Thus, the new way of producing colored ATO is an area of interest for research. 

In recent years, a new type of ethanol-based electrolyte for titanium anodizing was reported [20], but the possibilities and scope for using this electrolyte have not been fully described. The goal of this paper is to investigate the possibility of producing nanostructured colored anodic titanium oxide in new types of ethanol-based electrolyte. Compared to barrier-type anodic oxide, the nanotubular morphology of colored ATO presents an opportunity to increase future color saturation by sealing nanotubes and their coverings with an outer metallic layer. This paper presents a new way of fabricating nanostructured colored anodic titanium oxide in situ during titanium anodizing in an ethanol-based electrolyte.

## 2. Materials and Methods

The samples were prepared on the top of a 0.25 mm-thick titanium foil (99.5% pure Alfa Aesar, Kandel, Germany) with dimensions of 1.5 cm × 2 cm. Before anodization, samples were degreased by sonication in acetone and isopropanol; next, they were rinsed with deionized (DI) water and dried in the air. One-step titanium anodization was conducted in two electrode electrochemical cells with a platinum grid as a cathode and a titanium sample as an anode. Control of the applied voltage was realized by using an adjustable DC power supply (NDN, model DF1760SL5A, Warsaw, Poland). To maintain a constant temperature during anodization, a thermostat (HUBER, model MPC-K6, Offenburg, Germany) was used. A multimeter (RIGOL 3058E, Portland, OR, USA) was used to measure and transfer the registered current data to a computer. A new type of electrolyte for titanium anodization [20] was tested. An anodization electrolyte, an ethanol-based (99.8% pure POCH Basic, Gliwice, Poland) solution with different contents of DI water (2, 3.5, 5, or 10 vol%) and 0.3 M of ammonium fluoride NH4F (Sigma Aldrich, Poznań, Poland) was utilized. The one-step anodization process was performed at various voltages (30, 40, 50, and 60 V) at a constant temperature of 20 °C for 30 min.

The morphological characterization of anodic oxides was established by a scanning electron microscope, Quanta 3D FEG (FEI, Hillsboro, OR, USA), equipped with an EDS detector. Oxide thickness was determined from SEM cross-sectional images. The optical characterization of anodic oxides was determined using a UV-Vis spectrometer (Lambda 900, Perkin Elmer, Cracow, Poland). The calculation of theoretical color was based on the CIE 1931 color space system. 

## 3. Results

During titanium anodization, the current densities in the function of anodization time, called current curves, were registered (Figure 1 and Figure 2). It was shown that, depending on (1) the anodization voltage with constant electrolyte composition (Figure 1) and (2) the content of water in the ethanol-based electrolyte with a constant anodization voltage (Figure 2), the current density varied. The shape of collected current curves is unusual for Ti anodization, as described in detail in the first report of an ethanol-based electrolyte applied for titanium anodization [20]. 

Generally, the typical current curves registered during the anodization process consist of three previously well-described stages [49]. Stage I corresponds with compact layer formation on the Ti surface; stage II relates to initiation of irregular nanopore formation on the oxide layer; and stage III starts when the steady-state appears. This was observed in the growing of regular nanopores or a nanotube layer on the titanium oxide [49]. 

In this study, stage I was clearly visible, but stages II and III were difficult to identify. At the beginning of anodization, the current density increased; after reaching the maximum during stage Il it decreased slightly to stay at a comparable level for several hundred seconds. Thereafter, the current density decreased slowly. The duration of the process with a comparable level of current density was strongly affected by anodization voltage: in the range of given electrolyte composition, with an increase in anodization voltage, the decrease in current density occurred earlier and faster, as observed in the sharper shape of the curves in Figure 1. This phenomenon was most visible for processes conducted with the electrolyte containing 2 vol% of water (Figure 1a), but it also occurred for other electrolyte compositions. Anodization carried out in the electrolyte with 3.5 vol% of water exhibited slightly different behavior than anodization conducted in other electrolyte compositions (Figure 1b). In the electrolyte with applied voltages higher than 30 V, the current density dropped around 400 s and then started to increase. The shape of the current curves suggested that anodization in this electrolyte composition could be specific, as analyzed in detail elsewhere [20]. 

Figure 1 reveals that, with constant electrolyte composition, current density grows with increases in applied anodization voltage for all tested electrolytes. Moreover, a higher content of water in an electrolyte gives rise to a higher maximum current density registered at the beginning of the anodization process. For example, during Ti anodization carried out at 60 V, in the electrolyte with 2 vol% of water, the maximum current density was 2.5 × 10^−3^ A/cm^2^, while during anodization conducted in the electrolyte with 10 vol% of water, the maximum current density was almost three time higher at 7.2 × 10^−3^ A/cm^2^. Interestingly, for given anodization voltages, the effect of water content in electrolytes on registered current density is highly significant (Figure 2). For titanium anodization conducted in electrolytes with higher water content, greater current density at the beginning of the reaction was observed, especially for higher applied voltages (Figure 2). In addition, with constant anodization voltage, the decrease in current density after stage I of anodization occurs earlier and is sharper when the electrolyte water content is higher (Figure 2). This relationship is similar to the change in current density observed when anodization voltage increases in the range of one electrolyte composition (Figure 1). The results suggest that anodization conducted in ethanol-based electrolytes with higher water content is faster than in electrolytes with lower water content. As was shown for titanium anodization in the ethylene glycol-based electrolyte containing fluoride, it is possible to find optimum oxide growth conditions based on water content in an electrolyte solution [50]. 

Generally, water is consumed in reactions of anodic oxide growth [50], and more water in the electrolyte results in increasing electrolyte conductivity and nanotube length [51,52]. In this study, the presence of water in the electrolytes affected current density and therefore the reaction rate. This is reflected in the thickness of anodic oxides prepared in ethanol-based electrolytes with a higher water content (compare with data of ATO thickness).

The relationship between average current density, during the whole process upon applying anodizing voltage (Figure 3a), and electrolyte water content (Figure 3b), was analyzed. It was proven that average current density increases linearly with increases in applied anodizing voltage in the range of a given electrolyte composition for all tested contents of water in the ethanol-based electrolyte (Figure 3a). As the water content in the electrolyte increased, the angle of curve inclination decreased; for 5 and 10 vol% of water content in the ethanol-based electrolyte, the lines became almost parallel (Figure 3a).

The average current density in the function of electrolyte water content showed the biggest changes in the electrolyte with a smaller water addition (2 and 3.5 vol%, as shown in Figure 3b). Although the biggest maximum current density was registered for Ti anodizing in the electrolyte with 10 vol% of water, the average current density for processes conducted with the electrolytes with 5 and 10 vol% of water were very similar (Figure 3b). Further increases in the water content in the ethanol-based electrolyte did not affect the average current density achieved during titanium anodization; nevertheless, reactions conducted in the electrolyte with 10 vol% of water were faster.

The morphology of anodic oxides and the thickness of oxide layers were analyzed by SEM. In Figure 4, the top surface (a) and the cross-section view (b) of anodic titanium oxide generated at 50 V in the electrolyte with 10 vol% of water are shown as examples of obtained morphology. The ATO samples made in all tested conditions have similar, nanotubular morphologies with different morphological features. The values of oxide thickness and nanotube diameters are shown in Figure 5.

For anodic oxide generated in all tested compositions of the ethanol-based electrolyte, the tendency of nanotube diameter to change depending on applied anodizing voltage is not clear (Figure 5a). For ATO samples generated in electrolytes with higher water content (5–10 vol%), the value of nanotube diameters significantly decreases when applied voltage increases. In addition, the nanotube diameters of oxides generated in 2 vol% of water content decrease when applied potential increases, but this change is very slight. On the other hand, the diameters of nanotubes formed in the electrolyte with 3.5 vol% of water increase when the applied voltage increases. Further, the current density registered during Ti anodization carried out in the ethanol-based electrolyte with 3.5 vol% of water exhibits a different course than the current collected in the rest of the studied electrolyte compositions (compare Figure 1b). Observed differences in current density could affect nanotube diameter in resulting anodic oxides.

In Figure 5b, the change in oxide thickness, depending on anodizing conditions, is shown. The oxide thickness increased when anodizing voltage increased for the samples generated in the electrolytes with 3.5, 5, and 10 vol% of water content, which is typical behavior for oxides formed by anodization [8,9,17,18]. On the other hand, the samples formed in electrolytes with 2 vol% of water for all applied voltages show similar thicknesses of oxide layers, within error limits, and it is possible to notice a small downward trend of oxide layer thickness with increases in voltage. In this specific electrolyte composition, the oxide dissolution rate was probably higher than the oxide growth rate, resulting in a decrease in oxide layer thickness. 

In Figure 6, the calculated values of the CIE coordinates represented on the CIE diagram and the macroscopic view of samples are presented. It was shown that the ATO obtained in this study are characterized by different colors affected by fabrication conditions for ethanol-based electrolytes with 3.5, 5, and 10 vol% of water. This phenomenon could be related to different anodic oxide thicknesses and/or different morphologies of ATO samples. As discussed elsewhere [27], depending on the nanostructure’s morphology, the differing light wavelength is scattered and reflected from the oxide surface, and there is also interference with waves of the same frequency and phase, resulting in different colors. SEM analysis conducted on ATO prepared in this study revealed that, independent from fabrication conditions, the anodic oxides exhibit nanotubular morphology with different nanotube diameters. Moreover, thickness of oxides varied significantly depending on the anodizing conditions (applied voltage and electrolyte composition). Only the thickness of oxides generated in the electrolyte with 2 vol% of water were constant within the error range. In addition, the nanotube diameter of ATO formed in the electrolyte with 2 vol% of water is similar for all tested voltages. Further, all ATO samples formed in that electrolyte composition are similar and independent of applied voltage color, which may be referred to as navy blue (Figure 6a). Simultaneously, the color of samples generated in other electrolyte compositions vary with applied anodizing voltages (Figure 6b–d). An important finding is that the nanotube diameter and oxide thickness of e samples changed significantly with anodizing conditions. These observations suggest that the observed colors of anodic samples result from the synergy effect between nanotube diameter and oxide thickness. The research revealed that it is possible to control the color of ATO samples by changing anodizing conditions in ethanol-based electrolytes. 

To define the optical properties of the obtained anodic titanium oxides, the measurements of diffusive reflectance were carried out (Figure 7). Details of the maximum reflection wavelength for all tested samples are summarized in Table 1. The results of macroscopic color observation and reflectance measurements are consistent with calculated theoretical colors (Figure 6). 

On the reflectance plots registered for oxides formed in electrolytes with higher water content (containing 3.5, 5, and 10 vol% of water, as presented on Figure 7b–d), for each sample, the maximum reflection falls at a different wavelength, as wavelength increases with increases in anodizing applied voltage. What is significant is that the highest percentage values of reflectance (about 50%) were recorded for samples prepared in the ethanol-based electrolyte with 10 vol% of water. As shown in Figure 5, oxides generated in that electrolyte composition exhibit both the biggest nanotube diameters and oxide thicknesses in all tested samples. On the other hand, in the reflectance plots of ATO samples prepared in the ethanol-based electrolyte with 2 vol% of water, the maximum reflection for all four samples was approximately the same wavelength (Figure 7a and Table 1), which is consistent with macroscopic color observations (Figure 6a). Moreover, SEM analysis has shown that ATO formed in the electrolyte with 2 vol% of water have almost the same oxide layer thickness and very similar nanotube diameters (Figure 5); therefore, the reflectance plots are compatible with this analysis. Results suggest that the color of samples prepared in the present study was influenced by the thickness of the anodic oxide layer and nanotube diameter. 

For samples formed in specific anodizing conditions (Table 1), two clear maximum values of reflection in the visible light range were obtained. When these samples were viewed, depending on the observation angle, variable colors were observed, e.g., for the ATO sample generated in the ethanol-based electrolyte with 5 vol% of water at 40 V, the colors were yellow and purple. This observed phenomenon is known as iridescence, which is conditioned by the process of diffraction and interference of light on surfaces of spatially varied thickness.

## 4. Conclusions

Colored anodic titanium oxide was fabricated in ethanol-based electrolytes containing 0.3 M ammonium fluoride and various amounts of deionized water (2, 3.5, 5, or 10 vol%), at voltages that varied from 30 to 60 V and at a constant anodization temperature of 20 °C. It was found in the range of one electrolyte composition that the current density registered during titanium anodizing increased with an increase in applied potential. Moreover, with increased water content in the electrolyte, the current density recorded during anodizing at selected voltages increased. The resulting anodic oxides in all tested conditions had nanotubular morphology and thickness up to hundreds of nanometers. For electrolyte with 3.5, 5, and 10 vol% of water content, the anodic oxide layer thickness increased with increases in applied potential. Nanotube diameter for samples formed in electrolytes with 3.5 vol% of water increased with applied potential; whereas, for samples generated in electrolytes with 5 and 10 vol% of water, the opposite trend was observed: nanotube diameter decreased with applied potential increases. The ATO nanotube diameters and the oxide thickness of samples formed in electrolytes with 2 vol% of water content were independent of applied voltage and remained constant within the error range for all tested potentials. Moreover, the color of ATO formed in electrolytes with 2 vol% of water was blue and was independent from applied voltages, while the color of samples generated in other electrolyte compositions changed with applied voltages. For samples formed in selected conditions, iridescence was observed. It was proposed that the observed structural color of anodic titanium oxide was affected by anodic oxide layer thickness as well as by nanotube diameter. The research reveals that it is possible to control the color of ATO samples by changing anodizing conditions in ethanol-based electrolytes. The thin layers with brilliant colors may have potential in color displays, anti-counterfeiting technology, decoration, sensors, and/or optoelectronic applications. 

## Figures and Tables

**Figure 1 materials-14-06992-f001:**
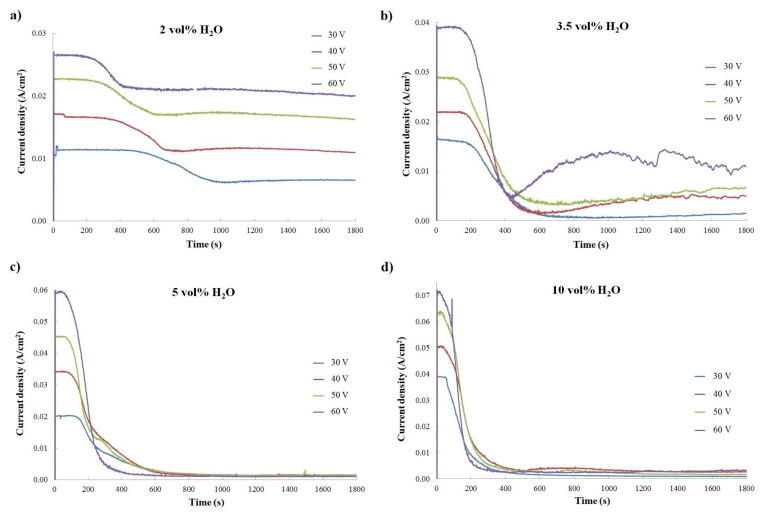
Current density in the function of anodization time showed, depending on anodization voltage for constant electrolyte composition: (**a**) 2 vol% of H_2_O; (**b**) 3.5 vol% of H_2_O; (**c**) 5 vol% of H_2_O; and (**d**) 10 vol% of H_2_O.

**Figure 2 materials-14-06992-f002:**
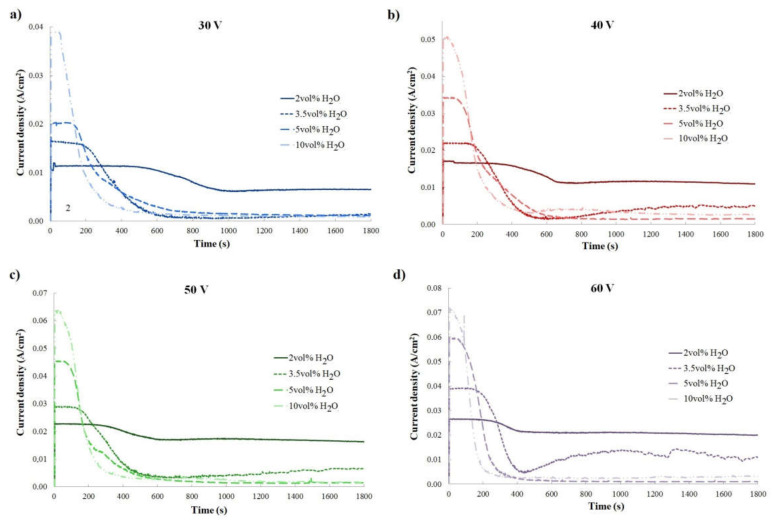
Current curves presented depending on content of water in ethanol-based electrolyte with constant anodization voltage: (**a**) 30 V; (**b**) 40 V; (**c**) 50 V; and (**d**) 60 V.

**Figure 3 materials-14-06992-f003:**
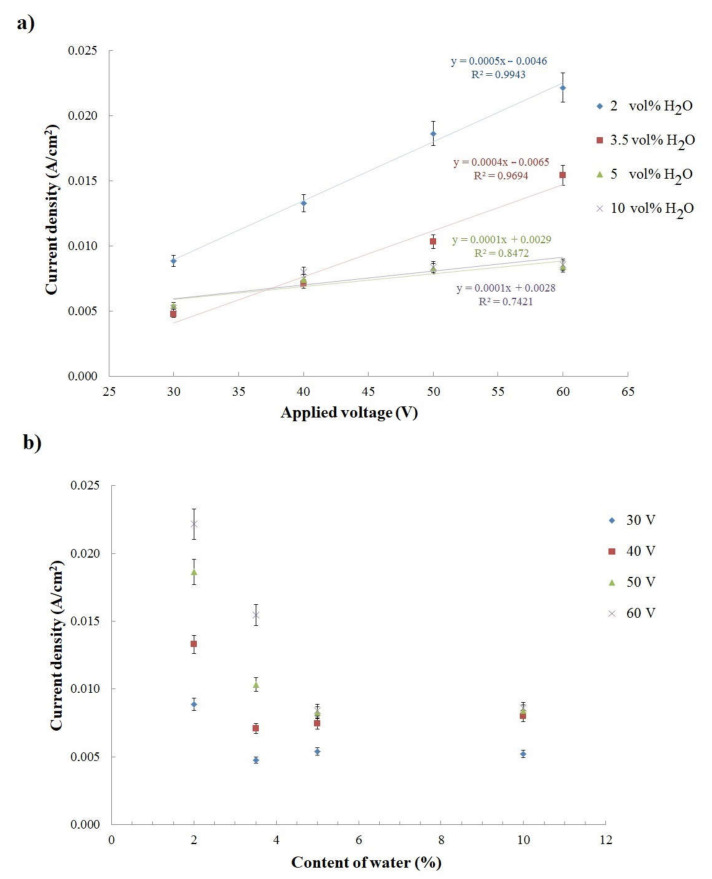
(**a**) Average current density in the function of applied anodizing voltage, and (**b**) water content in the electrolyte.

**Figure 4 materials-14-06992-f004:**
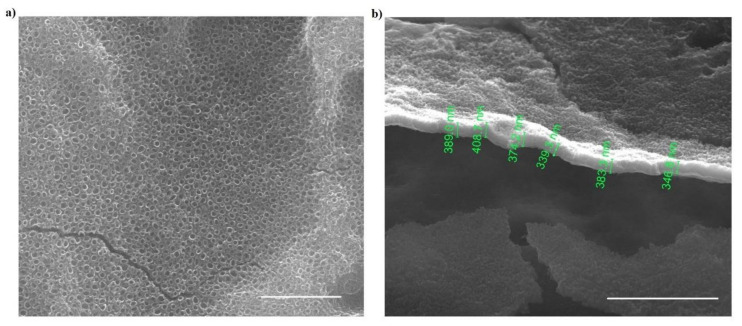
An example of morphology of anodic titanium oxides generated at 50 V in an electrolyte with 10 vol% of water: (**a**) top view, and (**b**) cross-section view. The bar corresponds to (**a**) 1 µm and (**b**) 3 µm of real distance.

**Figure 5 materials-14-06992-f005:**
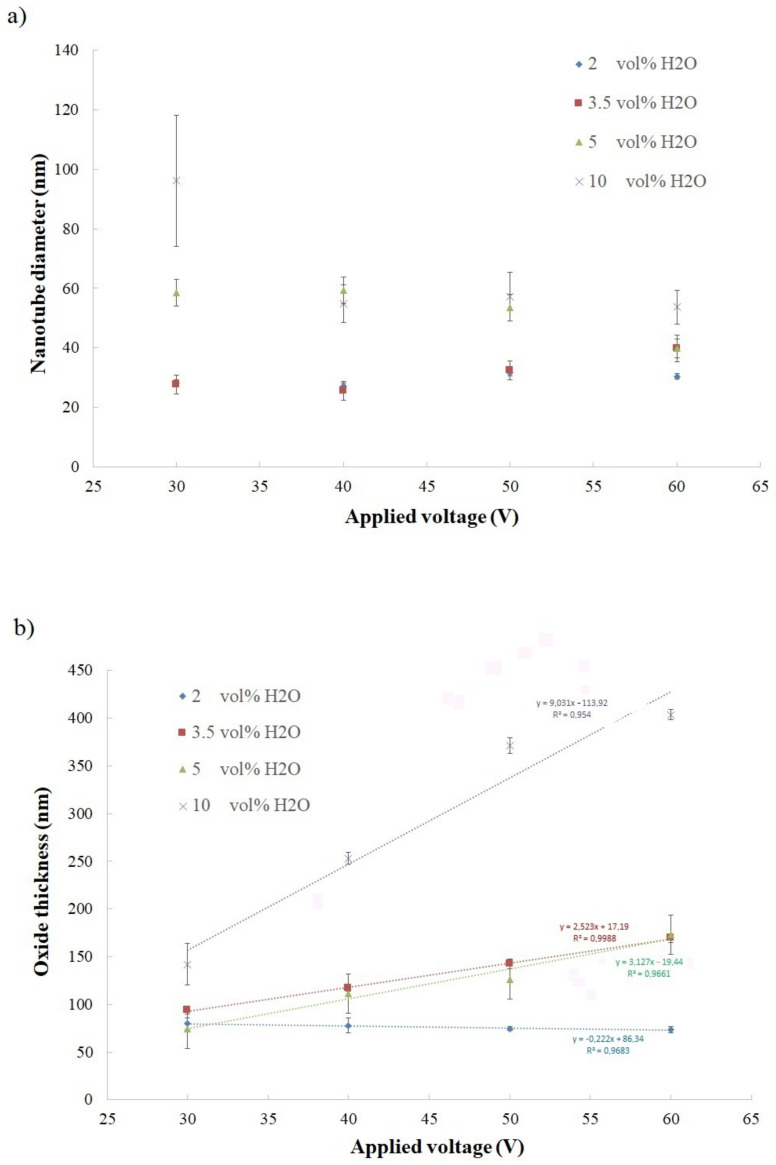
Nanotubes diameter in the function of applied voltage (**a**) and anodic titanium oxide thickness in the function of anodizing voltage (**b**).

**Figure 6 materials-14-06992-f006:**
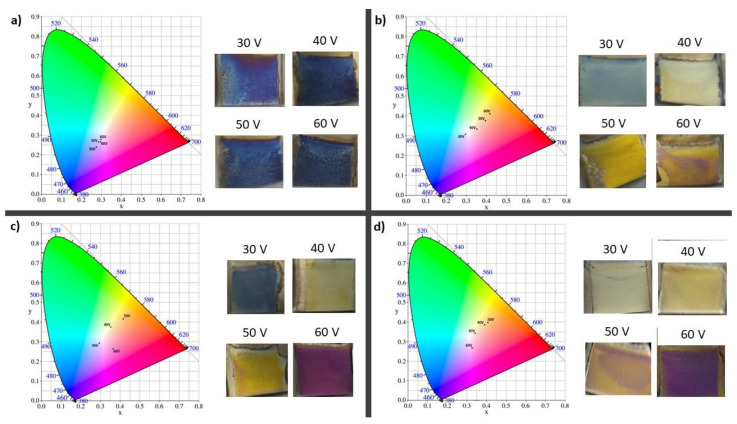
The calculated values of the CIE coordinates represented on the CIE diagram and the macroscopic view of titanium anodic oxide samples fabricated in electrolyte with (**a**) 2 vol% of H_2_O; (**b**) 3.5 vol% of H_2_O; (**c**) 5 vol% of H_2_O; and (**d**) 10 vol% of H_2_O.

**Figure 7 materials-14-06992-f007:**
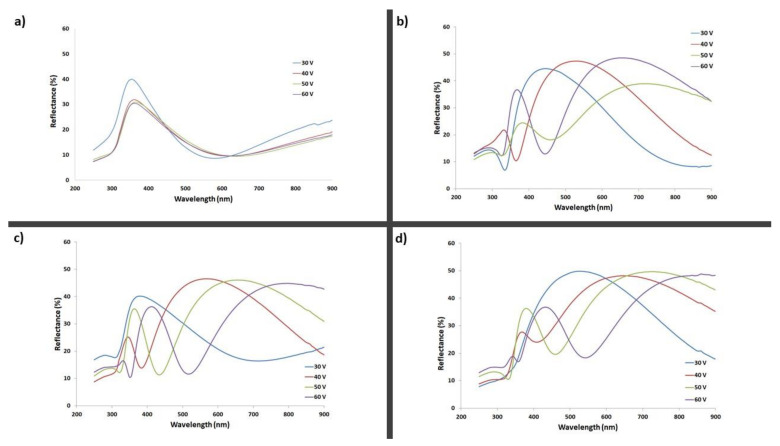
Diffusive reflectance plots for titanium anodic oxides formed in ethanol-based electrolytes with (**a**) 2 vol% of H_2_O; (**b**) 3.5 vol% of H_2_O; (**c**) 5 vol% of H_2_O; and (**d**) 10 vol% of H_2_O).

**Table 1 materials-14-06992-t001:** Wavelength values [nm] for the maximum reflectance obtained for samples generated in all tested electrolyte compositions.

	2 vol% H_2_O	3.5 vol% H_2_O	5 vol% H_2_O	10 vol% H_2_O
30 V	360	450	390	520
40 V	360	520	**550/380**	**620/380**
50 V	360	**620/380**	**640/390**	**700/390**
60 V	360	**700/390**	410	430

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
