# Peer review of "Morphological and Optical Characterization of Colored Nanotubular Anodic Titanium Oxide Made in an Ethanol-Based Electrolyte"

_materials, 2021, doi:10.3390/ma14226992_

Round 1
Reviewer 1 Report
This manuscript reports on the formation and properties of anodic titanium oxides from ethanol-based electrolyte with various additions of water. This manuscript is a continuation of a work published in ref 20, but it has some new findings, which are not properly addressed. Although this article contains some new information, I believe it should be completely re-written and resubmitted. At the same time, the manuscript suffers from a huge number of language/grammar flows.
- Since the research presented in this manuscript is focused on ATO I do not see a reason for writing over 40 lines in the introductory section about AAO. I believe that AAO should be just mentioned (having in mind there are some similarities with ATO) and ATO literature review should be the main part of the introduction. Also, authors should clearly explain the choice of the electrolyte and reasons for starting this work on Ti.
- Authors claim that ATO film thickness is responsible for the structural color. This contradicts Figs 5, 6 and Table 1. ATO coatings obtained at 60 V in electrolytes with 3.5 and 5 %vol of water have almost identical thickness (Fig5), while their visual appearance is different (Fig 6) as well as maximum reflectance values (Table 1). Authors should try to draw Tauc plots and extract absorption wavelength data which may be correlated to film thickness/visual appearance.
- This manuscript has an unusually high number of language/grammar flows. Some of the details (just for the first page) can be found below. These flaws make manuscript very hard to read and understand so I would propose a detailed and careful re-writing of the manuscript, possibly with the help of a native English speaker.
line 12: Please write °C.
line 13: layers instead of layer.
Line 14: calculating instead of calculate, resulting instead of resulted.
Line 17: ATO is mentioned for the first time.
Line 18: remain instead of remain.
Line 31: please write periodic micro- or nano- structures present on the surface.
Line 33: to produce instead of to produced.
Line 35: are instead of is
Line 123> bracket is not closed, samples were
……
Author Response
Dear Reviewer,
Thank you very much for your careful reviews of our manuscript. We find your comments as valuable and critical to our study. We have considered your comments and revised our manuscript carefully. Please find below our responds to your comments.
We believe that now our manuscript has been improved significantly. All change made in manuscript was highlighted on yellow.
Reviewer:
This manuscript reports on the formation and properties of anodic titanium oxides from ethanol-based electrolyte with various additions of water. This manuscript is a continuation of a work published in ref 20, but it has some new findings, which are not properly addressed. Although this article contains some new information, I believe it should be completely re-written and resubmitted. At the same time, the manuscript suffers from a huge number of language/grammar flows.
Remark 1: Since the research presented in this manuscript is focused on ATO I do not see a reason for writing over 40 lines in the introductory section about AAO. I believe that AAO should be just mentioned (having in mind there are some similarities with ATO) and ATO literature review should be the main part of the introduction. Also, authors should clearly explain the choice of the electrolyte and reasons for starting this work on Ti.
Respond:
Since AAO and ATO are most investigated and mainly used in industry anodic oxides, description of colored AAO in opinion of Authors are very important as a background of conducted research. What is important, a lot of similarities and reference between AAO and ATO can be found, so description of AAO is necessary. Despite of this we found that Reviewer remark as valuable, and therefore the part of manuscript focused on AAO was significantly shortened and we more focused on ATO. Moreover, the motivation of research in Introduction part was broadening.
Remark 2: Authors claim that ATO film thickness is responsible for the structural color. This contradicts Figs 5, 6 and Table 1. ATO coatings obtained at 60 V in electrolytes with 3.5 and 5 %vol of water have almost identical thickness (Fig5), while their visual appearance is different (Fig 6) as well as maximum reflectance values (Table 1).
Respond:
Thanks for this valuable remark. The samples made in electrolyte with 2 vol% of water have a similar thickness and nanotubes diameter. The samples made in electrolyte with other compositions have significantly different nanotubes diameters and oxide thickness, which affects the obtained color. At summary, the color of ATO depends on the oxide thickness and diameter of the nanotubes. Necessary improvements were done, and relevant sentences have been added to the manuscript.
Remark 3: Authors should try to draw Tauc plots and extract absorption wavelength data which may be correlated to film thickness/visual appearance.
Respond:
The reflectance tests were carried out with a constant resolution (5nm) in the range from 360nm to 830nm. Unfortunately, this range is too narrow to find the minimum and maximum energy distances of the absorbed photons for some samples (determination of energy gaps). The constant resolution of 5nm means that there is too little data for approximation. In that case, presentation of that research did not give additional knowledge to presented study, so Authors decide to not introduce Tauc plots to manuscript.
Remark 4: This manuscript has an unusually high number of language/grammar flows. Some of the details (just for the first page) can be found below. These flaws make manuscript very hard to read and understand so I would propose a detailed and careful re-writing of the manuscript, possibly with the help of a native English speaker.
line 12: Please write °C.
line 13: layers instead of layer.
Line 14: calculating instead of calculate, resulting instead of resulted.
Line 17: ATO is mentioned for the first time.
Line 18: remain instead of remain.
Line 31: please write periodic micro- or nano- structures present on the surface.
Line 33: to produce instead of to produced.
Line 35: are instead of is
Line 123> bracket is not closed, samples were
……
Thank you for your comments. Extensive language revision was done.
Reviewer 2 Report
The article is interesting and suitable for Materials, being focused on a new way of in-situ fabrication of colored anodic titanium oxide, during titanium anodizing in ethanol-based electrolyte.
The research flow is clear presented with all involved materials and methods.
Figure 1 should be presented on full wide page.
Same for Figure 2 and 7. In present form are too small to visualize.
Some recent references 2019-2021 should be provided to sustain the background of the research.
The conclusions are supported by the data highlighting the future use in color displays, anti-counterfeiting technology, decoration, sensors, or optoelectronic applications.
Author Response
Dear Reviewer,
Thank you very much for your kind reviews of our manuscript. We find your comments as valuable to our study. We have considered your comments and revised our manuscript carefully according to yours remarks. Please find below our responds to your comments.
We believe that now our manuscript has been improved significantly. All change made in manuscript was highlighted on yellow.
Reviewer: The article is interesting and suitable for Materials, being focused on a new way of in-situ fabrication of colored anodic titanium oxide, during titanium anodizing in ethanol-based electrolyte.
The research flow is clear presented with all involved materials and methods.
Remark 1: Figure 1 should be presented on full wide page.
Respond: The presentation of Figure 1 was change.
Remark 2: Same for Figure 2 and 7. In present form are too small to visualize.
Respond: The presentation of Figures 2 and 7 was change.
Remark 3: Some recent references 2019-2021 should be provided to sustain the background of the research.
Respond: More recent reference was added to manuscript text.
Reviewer: The conclusions are supported by the data highlighting the future use in color displays, anti-counterfeiting technology, decoration, sensors, or optoelectronic applications.
Thank you for your comments.
Reviewer 3 Report
- Please, check the English language: there are many typos in the text of the manuscript;
- Please, do not use abbreviations in the abstract;
- Introduction section: I would eliminate most of the reference to AAO since the goal of the manuscript is the ATO. Please, reduce a lot the introduction section regarding AAO and broad that one dedicated to ATO, also extending your comments on the interference colors of the barrier-type anodic oxides (see M.V. Diamanti research group papers);
- Please be careful when you write about anodization steps. Your anodization procedure is carried out with only one step but in the discussion of the results you wrote “1st step, 2nd step” and so on. Please add a brief paragraph on the various stages of the anodization procedure to have nanostructured oxides and, then, refer to stages instead of steps.
- Anodization curves: please, add a discussion on the shape of the current density curves. What is the effect of the water content? You wrote “with higher water content is more violent than in electrolyte with lowered water content. Probably more water in electrolyte resulted in increasing the number of electric charge carriers present in electrolyte and therefore affected reaction rate.” What do you mean with “violent”? It is not a scientific term…also the explanation regarding the number of electric charge carriers is rather weak. Please, add more details to the discussion.
- You definitely have nanotubes and not nanoporous titanium oxide. Please, remove porous from the title and write nanotubes.
- That the interference color of the anodic oxides change with the thickness is a very established result and I do not see any new contributions to the literature. What is the goal of your paper? Why is it so important the color and why you decided to make nanostructured ATO? The same result can be achieved with barrier-type anodic oxide. Please, broad your manuscript motivations and discuss what is the difference between what you obtain with nanostructured ATO and barrier-type ATO.
Author Response
Dear Reviewer,
Thank you very much for your careful reviews of our manuscript. We find your comments as valuable and critical to our study. We have considered your comments and revised our manuscript carefully. Please find below our responds to your comments.
We believe that now our manuscript has been improved significantly. All change made in manuscript was highlighted on yellow.
Reviewer:
Remark 1: Please, check the English language: there are many typos in the text of the manuscript;
Respond:
The extensive language revision was done.
Remark 2: Please, do not use abbreviations in the abstract;
Respond:
Thanks, the abbreviations were removed from the abstract.
Remark 3: Introduction section: I would eliminate most of the reference to AAO since the goal of the manuscript is the ATO. Please, reduce a lot the introduction section regarding AAO and broad that one dedicated to ATO, also extending your comments on the interference colors of the barrier-type anodic oxides (see M.V. Diamanti research group papers);
Respond:
Since AAO and ATO are most investigated and mainly used in industry anodic oxides, description of colored AAO in opinion of Authors are very important as a background of conducted research. What is important, a lot of similarities and reference between AAO and ATO can be found, so description of AAO is necessary. Despite of this we found that Reviewer remark as valuable, and therefore the part of manuscript focused on AAO was significantly shortened and we more focused on ATO. Moreover we add more suitable reference for fabrication of colored ATO.
Remark 4: Please be careful when you write about anodization steps. Your anodization procedure is carried out with only one step but in the discussion of the results you wrote “1st step, 2nd step” and so on. Please add a brief paragraph on the various stages of the anodization procedure to have nanostructured oxides and, then, refer to stages instead of steps.
Respond:
Thanks for that valuable remark. Part about various stages of anodization was add to the manuscript. In discussion part stage I, II and III of anodization was use.
Remark 5: Anodization curves: please, add a discussion on the shape of the current density curves. What is the effect of the water content? You wrote “with higher water content is more violent than in electrolyte with lowered water content. Probably more water in electrolyte resulted in increasing the number of electric charge carriers present in electrolyte and therefore affected reaction rate.” What do you mean with “violent”? It is not a scientific term…also the explanation regarding the number of electric charge carriers is rather weak. Please, add more details to the discussion.
Respond:
Thank you for that comment. We add more details and discussion about anodization curves. Also, the hastily used word has been changed to an appropriate scientific term
Remark 6: You definitely have nanotubes and not nanoporous titanium oxide. Please, remove porous from the title and write nanotubes.
Respond:
The title was change according to Reviewer suggestion.
Remark 7: That the interference color of the anodic oxides change with the thickness is a very established result and I do not see any new contributions to the literature. What is the goal of your paper? Why is it so important the color and why you decided to make nanostructured ATO? The same result can be achieved with barrier-type anodic oxide. Please, broad your manuscript motivations and discuss what is the difference between what you obtain with nanostructured ATO and barrier-type ATO.
Respond:
In that manuscript for the first time for fabrication of coloured ATO the new type of ethanol-based electrolyte was used and influence of anodizing conditions on resulted colour was describe in detail. Presented method of fabricate coloured oxide is no pigments used, what is very important to environment and people health. Moreover, the nanotubular morphology of coloured ATO presents an opportunity to futured increasing of colour saturation by sealing nanotubes and its covering by outer metallic layer. Sealing is impossible in the case of barrier-type oxide.
The manuscript motivation part was broadening and add to Introduction part.
……
Thank you for your comments.
Round 2
Reviewer 1 Report
The manuscript is acceptable in the amended form.
Reviewer 3 Report
Manuscript has been revised according to my suggestions